# The Mechanism Underlying the Amylose-Zein Complexation Process and the Stability of the Molecular Conformation of Amylose-Zein Complexes in Water Based on Molecular Dynamics Simulation

**DOI:** 10.3390/foods12071418

**Published:** 2023-03-27

**Authors:** Chaofan Wang, Na Ji, Lei Dai, Yang Qin, Rui Shi, Liu Xiong, Qingjie Sun

**Affiliations:** 1Key Laboratory of Food Processing Technology and Quality Control in Shandong Province, College of Food Science and Engineering, Shandong Agricultural University, Tai’an 271018, China; 2College of Food Science and Engineering, Qingdao Agricultural University, Qingdao 266109, China; 3Qingdao Special Food Research Institute, Qingdao 266109, China; 4Department of Food Science and Technology, College of Light Industry and Food Engineering, Nanjing Forestry University, Nanjing 210037, China

**Keywords:** molecular dynamics simulation, amylose, zein, complexation, conformational stability

## Abstract

The aim of this study was to employ molecular dynamics simulations to elucidate the mechanism involved in amylose–zein complexation and the stability of the molecular conformation of amylose–zein complexes in water at the atomic and molecular levels. The average root mean square deviation and radius of gyration were lower for amylose–zein complexes (1.11 nm and 1 nm, respectively) than for amylose (2.13 nm and 1.2 nm, respectively), suggesting a significantly higher conformational stability for amylose–zein complexes than for amylose in water. The results of radial distribution function, solvent-accessible surface area, and intramolecular and intermolecular hydrogen bonds revealed that the amylose–zein interaction inhibited water permeation into the amylose cavity, leading to enhanced conformational stabilities of the V-type helical structure of amylose and the amylose–zein complexes. Furthermore, the amylose in amylose–zein complexes displayed the thermodynamically stable ^4^C_1_ conformation. These findings can provide theoretical guidance in terms of the application of protein on starch processing aiming to improve the physicochemical and functional properties of starch (such as swelling capacity, pasting properties, and digestibility) for developing novel low-digestibility starch–protein products.

## 1. Introduction

Starch, a major food ingredient from cereals, plays an important role in supplying metabolic energy when consumed [1]. However, cooked starchy foods generally contain large amounts of rapidly digestible starch (RDS), which is quickly converted to glucose and usually causes dramatic increases in blood glucose levels. Consequently, starch is considered an important cause of several noncommunicable chronic diseases, such as obesity, type 2 diabetes, and cardiovascular and cerebrovascular diseases [2]. Therefore, the development of novel foods with starch of lower digestibility is receiving appreciable attention as a strategy for retarding rises in postprandial blood glucose levels and preventing excessive blood glucose concentrations. Several studies have demonstrated that interactions between starch and other food macromolecules (e.g., proteins) during food processing could reduce starch digestibility to a certain extent [3,4]. Clinical studies have indicated that exogenous proteins and their hydrolysates could mitigate starch digestion by forming complexes with starch, thereby lowering the glycemic response and ameliorating insulin resistance [5]. However, the mechanism underlying the starch–zein interaction has not yet been completely established.

The interactions between starch and protein are determined by a series of different interaction forces, including covalent and non-covalent interactions (e.g., hydrogen bonds, electrostatic forces, van der Waals forces, and hydrophobic actions) [6]. However, the contributions of these forces to the formation of starch–protein complexes and their conformational stability are unclear. Generally, the study of molecular conformations of chemical compounds and their interactions at the atomic or molecular levels requires the use of extremely sophisticated devices, such as nuclear magnetic resonance imaging, atomic force microscopy, and microscale thermophoresis. However, the wide application of these devices is limited by the expensive costs associated with testing and time-consuming pretreatment procedures [7]. Furthermore, for starch, due to its high molecular weight, correct measurement of the molecular conformation and the interactions between starch and other compounds requires that starch first be dissolved completely [8]. Starch is often dissolved in dimethyl sulfoxide (DMSO) rather than in water (the usual processing solvent for starchy foods) [9]. However, the molecular conformation of starch in water differs from that in DMSO [10], so determining the actual molecular conformation of starch and the interactions between starch and other compounds is difficult.

The motion state of different atoms and molecules and their interactions are now increasingly being studied using molecular dynamics simulation because of its low cost, high accuracy, and various functions [11]. Molecular dynamics simulation is a scientific calculation method that depends on Newtonian mechanics. It enables researchers to numerically integrate the motion equations of each particle in the whole system (ranging from hundreds to thousands of particles) in thousands of time steps (or more), where the motion paths of particles during the simulation represent the real motion trajectories [12]. The molecular configuration, thermal properties, and mechanical properties of a chemical compound in the simulation system can be obtained by analyzing trajectory files [7].

Recently, molecular dynamics simulation has been widely used to study the complexation processes and affinities between carbohydrates and other food components [13,14]. In a previous study, the stability of the intramolecular hydrogen bonds of amylose was increased by linoleic acid, which interacted with the amylose molecule, thereby confirming the greater stability of the V-type helical structure of amylose in amylose–linoleic acid complexes than in the single-strand system [14]. Chen et al. (2019) [13] successfully identified the intermolecular hydrogen bonds between hydroxyl groups of the amylose molecule and the amino (carboxyl) groups of the zein molecule.

When starchy foods are hot-processed in water, the crystalline regions of the starch molecules are penetrated by water molecules and are gradually loosened, and amylose molecules effuse from the starch granules [15]. In the presence of hydrophobic ligands (e.g., zein molecules), the interactions between amylose and these ligands at room temperature can increase the conformational stability of the amylose molecules [13]. Our research team recently fabricated core-shell starch-zein microparticles with high slowly digestible starch and resistant starch contents [16]. The core materials were swollen corn starch granules, that were prepared by heating in excess of water. At room temperature, the preformed swollen corn starch granule suspension was dripped into the zein ethanolic solution to prepare core-shell starch–zein microparticles [16]. However, further investigation is required to determine the accurate physicochemical mechanisms involved.

In the present study, to explore the mechanism underlying the starch–zein interaction in water at room temperature, we simulated the complexation process and motion trajectories of an amylose–zein complex at room temperature. In total, three independent simulation models with different initial values were run for amylose–zein systems (designated amylose–zein complex 1, 2, and 3) to assess the reproducibility of the simulations. We also studied the conformational stability of the amylose molecule with a V-type helical structure in the presence and absence of a zein molecule. Our goal was to fill in the research gaps by addressing several questions left unanswered by previous studies: (1) How does the amylose molecule interact with the zein molecule, and how do the amylose–zein complexes maintain conformational stability in water? (2) Which forces contribute more to the formation of starch–zein complexes and their conformational stability in water? (3) How do intramolecular and intermolecular hydrogen bonds change during the formation of amylose–zein complexes? (4) What is the thermodynamically stable configuration of amylose–zein complexes? A previous study suggested that the conformational stability of the amylose molecule affected the starch physicochemical properties such as swelling capacity and gelatinization temperatures [17], which have been proven to affect starch digestibility [16,18]. Thus, revealing the answers to the above questions can provide theoretical guidance in terms of the application of zein on starch-processing aiming to improve the physicochemical and functional properties of starch (such as swelling capacity, pasting properties, and digestibility) for developing novel low-digestibility starch–zein products.

## 2. Materials and Methods

### 2.1. Materials

The amylose model, modeled by Material Studio 8.0 (Accelrys Software Inc., San Diego, CA, USA), was composed of 26 glucose residues with ^4^C_1_ conformation, per V-type left-handed helix that consisted of 6 glucose residues. The zein molecule was modeled by Modeller 9.25 (Sali Lab., UCSF, San Francisco, CA, USA) [19].

### 2.2. Force Field

All molecular dynamics simulations were carried out using GROMACS 5.1.4 (SciLife Lab., Stockholm, Sweden), where the glucose force field was modified from the OPLS-AA-SEI force field for the amylose molecule [20] and the transferable intermolecular potential 3-point (TIP3P) water model was used for the water molecules [21].

### 2.3. Molecular Dynamics Simulation Parameters

For all simulations, the leap-frog scheme was applied to integrate the equations of motion with a time step length of 2 fs [22]. The linear constraint solver (LINCS) algorithm was used to constrain all bond lengths [23]. The V-rescale thermostat was applied to maintain a reference temperature of 300 K with relaxation times of 1 ps [24], and the reference pressure was coupled isotropically by employing a Parrinello–Rahman barostat with a coupling constant of 2 ps to maintain pressure at 1 bar [25]. Both short-range electrostatic and van der Waals cutoff distances were 1 nm [26], and long-range electrostatic interactions were calculated by the Particle Mesh Ewald method [27]. Molecular dynamics simulations were run with an equilibrium of 2 ns in both the canonical ensemble (NVT) and the isobaric-isothermal ensemble (NPT). Production simulations were run for 1000 ns in both the amylose system and the amylose–zein system after reaching equilibrium. The amylose molecule exhibited a V-type helical structure at 0 ns in both the amylose and amylose–zein systems. The criteria used for determining the existence of a hydrogen bond were a distance between the hydrogen bond acceptor atoms and donor atoms within 0.35 nm and an angle between the hydrogen bond acceptor atoms and donor atoms within 30°. The simulation files were extracted every 0.01 ns to calculate the number of intramolecular and intermolecular hydrogen bonds, root mean square deviation (RMSD), radius of gyration (R_g_), root mean square fluctuation (RMSF), end-to-end distance and atom distance, radial distribution functions (RDF), solvent accessible surface area (SASA), principal component analysis (PCA), and the distribution proportion of the ^1^C_4_ chair conformation.

### 2.4. Statistical Analysis

All data were statistically evaluated and presented as mean values ± standard deviations (SD) of three replicates for each sample using SPSS 22.0 (IBM Corporation, Chicago, IL, USA).

## 3. Results and Discussion

### 3.1. Conformational Transitions of the Amylose Molecule and Amylose–Zein Complexes in Water

Figure 1A,B display the conformational transitions of an amylose molecule and amylose–zein complexes in water at different times between 100 ns and 1000 ns, respectively. The trajectories of the water molecules were concealed to allow distinct observations of the trajectories and conformational transitions of the amylose molecules and amylose–zein complexes. The amylose molecules exhibited the V-type helical structure at 0 ns in both the single-strand system and the complexes. At 0 ns, the amylose and zein molecules were randomly distributed in a cubic water box. As the simulation time increased, the V-type helical structure of a single amylose molecule in water was gradually loosened and then completely extended at around 80 ns. A similar result was obtained by Cheng et al. (2018), who showed that the V-type helical structure of a single amylose molecule unfolded completely in water before 100 ns [14]. In the presence of zein, the zein molecules gradually came into contact with the middle sections of the amylose molecule, and the interactive process started. For the amylose chains in complexes, the V-type helical structure of the two ends loosened at around 400 ns, while the middle parts were preserved until 1000 ns, indicating that the presence of zein preserved a partial V-type helical structure.

In a previous study, the amylose molecules in amylose–ligand complexes exhibit a left-handed V-helical structure [28], where each helix might contain six, seven, or eight glucose units, designated as V6-, V7-, or V8-amylose inclusion complexes, respectively [29]. The amylose molecule is reported to interact with the zein molecule in water to form V6-amylose inclusion complexes [13], which was also confirmed by our research results.

### 3.2. Intramolecular and Intermolecular Hydrogen Bonds

The presence of hydrogen bonds plays a crucial role in maintaining the stability of the helical conformation. Therefore, the intramolecular hydrogen bonds of the amylose molecule were determined in both the single-strand system and the three amylose–zein complexes with different initial values (Figure 2A). For amylose alone, the number of amylose–amylose hydrogen bonds ranged from 0–10 and accounted for approximately 95% of the total. By contrast, for an amylose molecule interacting with a zein molecule, the number of amylose–amylose hydrogen bonds was much higher, suggesting that the zein molecule maintained the stability of the amylose helical structure.

Generally, the V-type helical structure of amylose is easily penetrated by water molecules in the single-strand system because of the small size and strong dipolarity of the water molecules. Therefore, the amylose–amylose hydrogen bonds are gradually replaced by amylose–water hydrogen bonds, and this replacement destroys the V-type helical structure of the amylose molecule [29]. However, the polar amino acids of zein molecules (particularly the charged polar amino acids) can interact with water molecules by forming competitive intermolecular hydrogen bonds, thereby suppressing the penetration of water molecules into the V-type helical structure of the amylose molecule [30].

One prime reason why an amylose molecule can form a complex with a zein molecule is the intermolecular hydrogen bonds that form between the hydroxyl groups of the amylose molecule and the amino (carboxyl) groups of the zein molecule [13]. Therefore, the distribution of the numbers of amylose–zein hydrogen bonds was determined for the three amylose–zein complexes with different initial values (Figure 2B). The amylose–zein hydrogen bonds ranging from 10–20 and over 20 accounted for over 95% of the total. Previous studies have shown that the polar amino acids of the zein molecule can interact with starch chains [16]. In particular, the partial amylose–water hydrogen bonds were replaced by amylose–zein hydrogen bonds due to the presence of charged polar amino acids [30].

As shown in Figure 2C, for an amylose chain containing 6 glucose units per helix, the intramolecular hydrogen bonds can be divided into O_2_-O_3_ single bonds between two adjacent glucose units and O_6_-O_2_/O_3_ single bonds between residue x and residue x+6 [29]. The V-type helical structure is dominated by the number and stability of O_6_-O_2_/O_3_ single bonds between residue x and residue x+6 [29]. The changes in the intramolecular hydrogen bonds in the amylose molecule in both the single-strand system and the three amylose–zein complexes with time are shown in Table 1 and Table 2. At 0 ns, the number of O_2_-O_3_ single bonds between two adjacent glucose units in amylose in both the single-strand system and the three amylose–zein complexes was 23 (Table 1). For the amylose molecule in the single-strand system, the number of O_2_-O_3_ single bonds between two adjacent glucose units decreased rapidly to 5.42 in the first 200 ns. By contrast, for the amylose molecule in the three amylose–zein complexes, the number of O_2_-O_3_ single bonds was only reduced to approximately 15.76 in the first 200 ns, and finally dropped to approximately 11.16 at 1000 ns. The number of O_6_-O_2_/O_3_ single bonds between residue x and residue x+6 of the amylose molecule in both the single-strand system and three amylose–zein complexes was 15 at 0 ns (Table 2). For the amylose molecule in the single-strand system, this number was reduced to 0 at the first 4 ns, which is consistent with the results for an amylose molecule in water obtained by Tusch et al. (2011) [10].

These results indicate that the V-type helical structure of the amylose molecule in the single-strand system was degraded quickly. This was especially the case for the O_6_-O_2_/O_3_ single bond between residue x and residue x+6, in which the degradation was the major cause of loosening of the amylose V-type helical structure. This is because the intramolecular hydrogen bonds in the amylose molecule are gradually replaced by intermolecular hydrogen bonds between amylose and water molecules [29]. For the amylose chain in the complexes, the presence of zein partially preserves the intramolecular hydrogen bonds of the amylose molecule. Therefore, there is greater conformational stability for the amylose molecule in the amylose–zein complexes than in the single-strand system.

### 3.3. Root Mean Square Deviation (RMSD)

RMSD is an effective tool for evaluating the average atomic displacement of a conformation at an instant of the simulation relative to a reference frame. The reference frame is usually the first frame of the simulation, or another different conformation at that instant. The stabilization of the RMSD value suggests that a stable balance has been achieved for a conformation or between two different conformations [31]. The RMSD values for the amylose molecule and the three amylose–zein complexes were calculated and are illustrated in Figure 3A. Similar to the results obtained by Chen et al. (2019) [13], the RMSD value of the amylose molecule showed a rapid increase within the first 10 ns, and then showed significant fluctuations until 100 ns, but ultimately maintained a relatively stable state. By contrast, the RMSD values of the three amylose–zein complexes achieved an equilibrium state faster after the first 10 ns, indicating that the simulation system was practical and stable. Figure 3A shows minor differences in the RMSDs of the three amylose–zein complexes, suggesting that these simulation systems were reproducible. The amylose molecule showed conformational instability, with an average RMSD of about 2.13 nm, whereas the three amylose–zein complexes had average RMSD values of 1.11 nm, indicating that the complexes achieved a relatively stable balance.

### 3.4. Radius of Gyration (R_g_)

R_g_ is an important indicator used to determine the distribution of an object relative to its initial position [32]. Figure 3B shows the R_g_ distributions of each trajectory of the amylose molecule and the three amylose–zein complexes in water. The R_g_ values for the amylose molecule are distributed in a large range, from 0.75 nm to 2.0 nm, similar to the results obtained by Cheng et al. (2018) [14]. By contrast, the R_g_ distribution of the three amylose–zein complexes ranged from 0.8 nm to 1.7 nm, suggesting a smaller range of fluctuations for the R_g_ values of the three amylose–zein complexes. Figure 3C shows the R_g_ of amylose molecule and three amylose–zein complexes in water. The average R_g_ values were lower for the three amylose–zein complexes (approximately 1.23 nm) than for the amylose molecule (approximately 1.54 nm), suggesting that amylose–zein complexes showed smaller fluctuations and higher conformational stability due to the interaction between amylose and zein.

### 3.5. Root Mean Square Fluctuation (RMSF)

The fluctuation degree of an object relative to a reference frame can be measured using the RMSF parameter. Figure 3D shows the RMSF value of each glucose residue of the amylose molecule of a single-strand system and the average RMSF value of each glucose residue in the three amylose–zein complexes, taking the O_5_ atom of the amylose molecule as a reference frame. For the amylose molecule and the amylose–zein complexes, the RMSF value was higher for the two ends of the amylose molecule (residue 1 and residue 26) than for the other residues of the amylose molecule, indicating a greater flexibility in the residues at the two ends of the amylose molecule. Overall, the conformation is more stable for the amylose–zein complex than for the amylose molecule, as confirmed by the lower RMSF values. Furthermore, the RMSF value fluctuations of the middle residues of the amylose molecule were lower in the amylose–zein complexes than in the single-strand system, confirming that the zein molecule interacted with the middle parts of the amylose molecule and further increased the conformational stability in that middle area. This result is consistent with the evidence presented in Figure 1.

### 3.6. End-to-End Distance and Atom Distance

The probability distribution of the distances between the two ends of an amylose molecule (residue 1 and residue 26) in both the single-strand system and the three amylose–zein complexes are illustrated in Figure 4. The distance between the two ends was slightly longer in the three amylose–zein complexes than in the single-strand system. This difference probably reflects the steric hindrance between the amylose molecule and the inclusion ligand (zein) [4]. The amylose molecule exhibited smaller fluctuations in complexes than in the single-strand system due to the interaction between amylose and zein, as indicated by the more concentrated probability distribution of the end-to-end distances of the amylose molecule in the complexes than in the single-strand system. This result is in agreement with the evidence shown in Figure 3B.

Previous work has shown that a loosened amylose chain can reform a V-type helical conformation when it interacts with a zein molecule [13], and that the amylose chain contains 6 glucose units per helix [29]. In Figure 5, the distance between the O_3_ atom of residue x and the C_5_ atom of residue x+3 was used to indicate the atom distance, as displayed in yellow in Figure 5. For both the amylose molecule (Figure 6A) and the three amylose–zein complexes (Figure 6B–D), the distances between the atoms are larger at the two ends than at the middle parts. The average atom distance was longer in the single-strand system (1.28 nm) than in the three amylose–zein complexes (1.19 nm). More importantly, the fluctuation in the atom distance and the atom distance of the middle parts of the amylose molecule were lower in the three amylose–zein complexes than in the single-strand system. This probably reflects that the interaction between the zein molecule and the amylose molecule increases the conformational stability of the amylose molecule and preserves partial V-type helical configurations of the middle parts of the amylose molecule. These results are consistent with the evidence shown in Figure 3D.

### 3.7. Radial Distribution Functions (RDF)

The RDF is applied to describe the density of water molecules within a certain distance (r) from a reference particle. In the case of amylose, it indicates the density changes in water around an amylose molecule with a change in distance from the amylose molecule [14]. The RDF of the amylose molecule in the single-strand system and the average RDFs of the amylose molecules in the three amylose–zein complexes are displayed in Figure 7. The peak at 0.275 nm represents the average bond length between the oxygen atom of the amylose molecule and the nearest water molecules, suggesting that a water shell has formed around the amylose molecule. For an amylose molecule in the single-strand system or the complexes, with r increased, an increase in r led to a sharp decrease in the g(r) values, followed by a dramatic rise at around 0.37 nm, and finally became steady at around 2.5 nm. The inset in Figure 7 shows that the g(r) value at 0.275 nm is lower for the amylose–zein complexes than for amylose in the single-strand system. This suggests that the interaction between the amylose and zein molecule inhibited water permeation into the amylose cavity; therefore, fewer water molecules formed a water shell around the amylose. By contrast, for the amylose molecule in the single-strand system, water molecules were more likely to permeate into the V-type helical structure of the amylose molecule, thereby destroying the intramolecular hydrogen bonds of the amylose molecule and leading to the loosening of the V-type helical structure.

### 3.8. Solvent Accessible Surface Area (SASA)

The SASA represents the surface area of a reference particle that can be accessed by solvent molecules [33]. The contact areas between the water molecule and the amylose molecule in both the single-strand system and three amylose–zein complexes are shown in Figure 7. For an amylose molecule in the single-strand system, the SASA value gradually increased within the first 80 ns, and then became steady until 1000 ns. This indicated that water molecules gradually permeated into the V-type helical structure of the amylose molecule at the first 80 ns and that the intermolecular hydrogen bonds between the amylose molecule and the water molecules gradually replaced the intramolecular hydrogen bonds of the amylose molecule. This induced a loosening of the helical structure of the amylose molecule, thereby increasing the surface area of the amylose molecule that could be accessed by water molecules. After 80 ns, the V-type helical structure of the amylose molecule was fully extended, and the contact area between the water molecules and the amylose molecule in the single-strand system reached a plateau that showed a small variation. Conversely, the SASA values for an amylose molecule were consistently lower in the three amylose–zein complexes than in the single-strand system. This suggests that the formation of amylose–zein complexes protected the amylose molecule from water molecules; therefore, the surface area of the amylose molecule that could be in contact with water molecules was smaller in the complexes than in the single-strand system. The smaller contact area between the water molecules and the amylose molecule in the complexes might also be induced by interactions between the charged polar amino acids of the zein molecule and the surrounding water molecules by competitive intermolecular hydrogen bonds [30].

### 3.9. Principal Component Analysis (PCA)

PCA is generally used to visually analyze whether one or more parameter combinations exist that can describe the conformational differences between an amylose molecule and amylose–zein complexes [34]. PCA can describe the major fluctuations in the amylose molecule and amylose–zein complexes during a simulation. The planar projection of the amylose molecule and the three amylose–zein complexes determined by the two major eigenvectors of the matrix-positional covariance matrix based on PCA are shown in Figure 8. The color variations from purple and blue to red represent the levels of energy from low to high. As shown in Figure 8A, the low-energy regions (purple and blue regions) of the amylose molecule only accounted for a small fraction of the plane determined by two eigenvectors, while the high-energy regions (red regions) of the amylose molecule occupied most of the plane determined by two eigenvectors. This indicated that the amylose molecule in the single-strand system displays an unstable conformation, most of which was at a high energy level. As shown in Figure 8B–D, the low-energy regions (purple and blue regions) of the three amylose–zein complexes accounted for a large part of the plane determined by two eigenvectors, suggesting that the conformations of amylose–zein complexes differed from that of the amylose molecule and that most of the amylose–zein complexes were at low energy levels. Furthermore, the distribution of energy regions were more concentrated for the amylose–zein complexes than the amylose molecule, meaning that the degrees of conformational variation were lower for the amylose–zein complexes than the amylose molecule. The low-energy regions (purple and blue regions) of the amylose–zein complexes were located at the middle parts of the planar projection determined by two principal eigenvectors, while the high-energy regions (red regions) of the amylose–zein complexes were located at the outside parts of the planar projection determined by two principal eigenvectors. This suggested that the conformational stability was higher for the middle parts of the amylose–zein complexes than for the two ends due to the interaction between the middle parts of the amylose molecule and the zein molecule. This supports the results obtained in Figure 1. A similar result was reported by Xu et al. (2015), who showed that the energy was lower for the middle parts of the amylose molecule than for the two ends of the amylose molecule when it interacted with a protein molecule [35].

### 3.10. The Distribution of the Proportion of the ^1^C_4_ Chair Conformation

The conformation of the six-membered ring of glucose residues can be categorized into two types: the chair conformation and the boat conformation. The conformational stability is higher for the chair conformation than for the boat conformation due to the lower energy of the chair conformation [36]. Figure 9A shows the two chair conformations of a six-membered glucose ring: the ^4^C_1_ (left) and the ^1^C_4_ (right). The angle θ represents the angle between the six-membered ring and the reference plane, which can usually distinguish the ^4^C_1_ conformation from the ^1^C_4_ conformation. When θ ranges between 0° and 60°, the six-membered rings of the amylose molecule exhibit the ^4^C_1_ chair conformation, whereas when θ is larger than 120°, the six-membered ring assumes the ^1^C_4_ chair conformation. The energy level is lower for the ^4^C_1_ chair conformation than for the ^1^C_4_ chair conformation; therefore, the conformational stability is higher for the ^4^C_1_ than for the ^1^C_4_ chair conformation [37].

The initial conformation of all six-membered rings of the glucose residues of the amylose molecule is the ^4^C_1_ chair conformation in both the single-strand system and the three amylose–zein complexes. The proportion of ^1^C_4_ chair conformation of glucose residues of the amylose molecule was measured in both the single-strand system and three amylose–zein complexes to determine the conformational stability, as shown in Figure 9B–E. The proportion in the ^1^C_4_ chair conformation was very low (<20%) for all six-membered rings of glucose in both the amylose molecule and the amylose–zein complexes, suggesting that most of these six-membered rings were in the ^4^C_1_ chair conformation. Similarly, a molecular dynamics simulation conducted by Casset et al. (1995) using the parameters appropriate for protein–carbohydrate interactions found that the ^4^C_1_ chair conformation was the major conformation of the six-membered glucose rings in a complex between amylose and porcine pancreatic α-amylase [38]. In general, the proportion of the ^1^C_4_ chair conformation in the amylose molecule is lower in the amylose–zein complexes than in the single-strand system. This reveals that the amylose molecule in the amylose–zein complexes have more glucose residues in the ^4^C_1_ chair conformation and therefore higher conformational stability in the amylose–zein complexes than in the single-strand system. More importantly, the proportion of glucose residues in the ^1^C_4_ chair conformation in both the complexes and the single-stranded system was higher in the two ends of the amylose molecule than in other regions of the amylose molecule. Conversely, the proportion of glucose residues in the ^1^C_4_ chair conformation in both the single-strand system and the amylose–zein complexes was lower in the middle of the amylose molecule than in other regions of the amylose molecule. Furthermore, the proportion of glucose residues in the ^1^C_4_ chair conformation of the middle parts of the amylose molecule was lower in the amylose–zein complexes than in the single-strand system. This indicates that the middle parts of the amylose molecule interacted with the zein molecule; therefore, the conformational stability of the middle parts of the amylose molecule increased, while the two ends of the amylose molecule remained flexible. Consequently, the V-type helical structure of the two ends of the amylose was more easily loosened and randomly coiled in water. This result is in agreement with the evidence shown in Figure 3D.

## 4. Conclusions

In this study, the mechanism underlying the amylose–zein complexation process and the stability of the molecular conformation of amylose–zein complexes in water were investigated by using molecular dynamics simulation. The middle parts of the amylose molecule were found to interact with the zein molecule, and this interaction was the dominant factor leading to the preservation of the partial V-type helical configuration of the amylose molecule in amylose–zein complexes. The conformational stability was significantly higher for the amylose–zein complexes than for a single amylose molecule in water. The results obtained for intramolecular and intermolecular hydrogen bonds, RDF, and SASA collectively showed that the amylose–zein interaction provided some protection to the V-type helical structure of the amylose molecule from water molecules, thereby improving the conformational stability of the V-type helical structure of the amylose molecule and the amylose–zein complexes. In addition, the highly thermodynamically stable ^4^C_1_ conformation was the major glucose conformation of amylose molecule in amylose–zein complexes. Our findings can serve as a complement to previous experimental results and as theoretical guidance for the application of proteins on starch-processing aiming to improve the physicochemical and functional properties of starch (such as swelling capacity, pasting properties, and digestibility) for developing novel low-digestibility starch–protein products.

## Figures and Tables

**Figure 1 foods-12-01418-f001:**
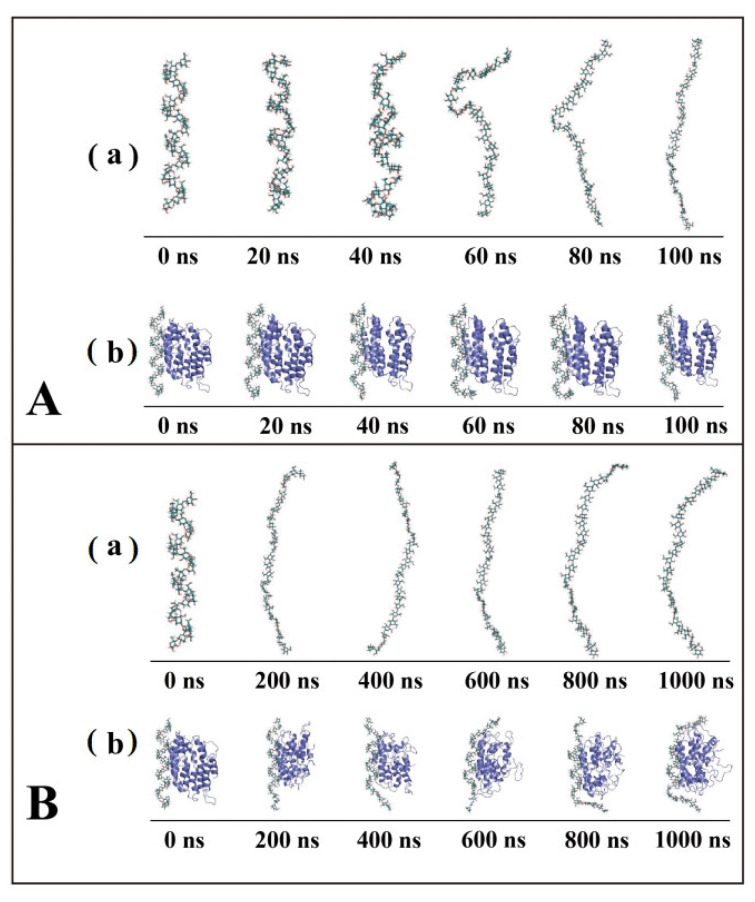
Conformational changes and snapshots of amylose molecule (**a**) and amylose–zein complexes (**b**) in water at different time intervals between 100 ns (**A**) and 1000 ns (**B**).

**Figure 2 foods-12-01418-f002:**
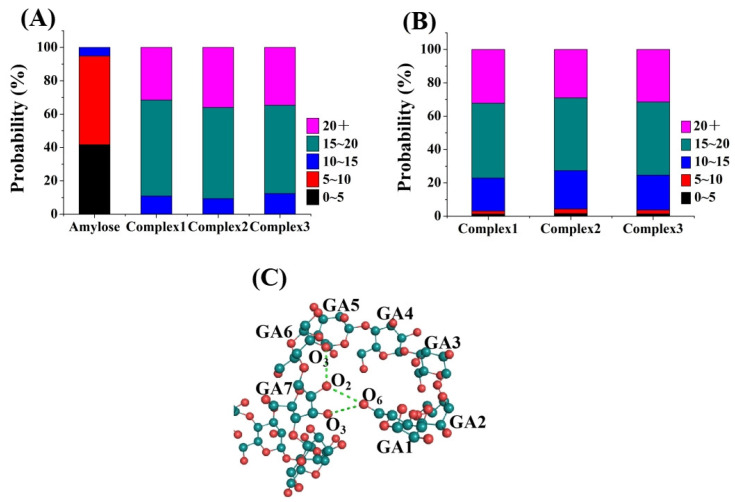
(**A**) Percentages of intramolecular hydrogen bonds number of amylose molecule in single-strand system and three amylose–zein complexes with different initial values (the number of intramolecular hydrogen bonds are shown in the figure legend); (**B**) Percentages of intermolecular hydrogen bonds number between amylose molecule and zein molecule in three amylose–zein complexes with different initial values (the number of intermolecular hydrogen bonds are shown in the figure legend); (**C**) O_2_-O_3_ single bond between two adjacent glucose residues (GR) and O_6_-O_2_/O_3_ single bond between residue x and residue x+6.

**Figure 3 foods-12-01418-f003:**
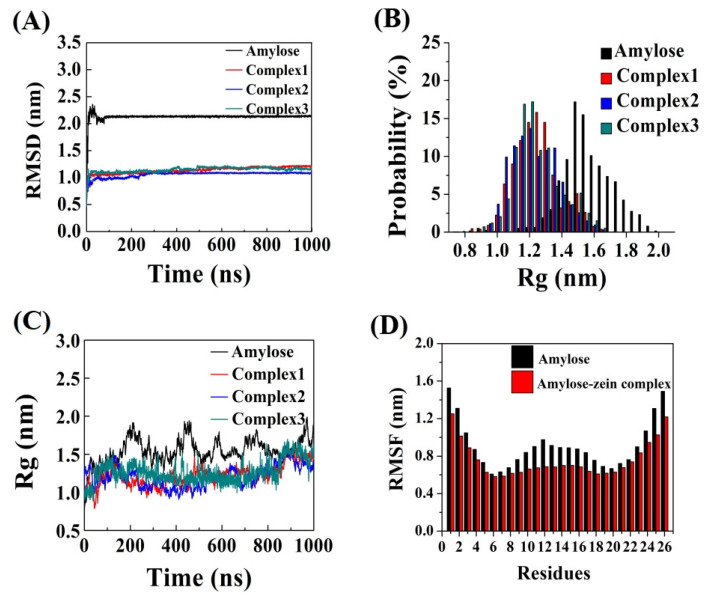
(**A**) Root mean square deviation (RMSD) of amylose molecule (black line) and three amylose–zein complexes with different initial values (red, blue and green lines); (**B**) Distribution of radius of gyration (R_g_) of amylose molecule (black bars) and three amylose–zein complexes with different initial values (red, blue and green bars); (**C**) Radius of gyration (R_g_) of amylose molecule (black line) and three amylose–zein complexes with different initial values (red, blue and green lines); (**D**) Root mean square fluctuation (RMSF) values of each glucose residue of amylose molecule in single-strand system (black bars) and the average RMSF values of each glucose residue of amylose molecule in three different amylose–zein complexes with different initial values (red bars).

**Figure 4 foods-12-01418-f004:**
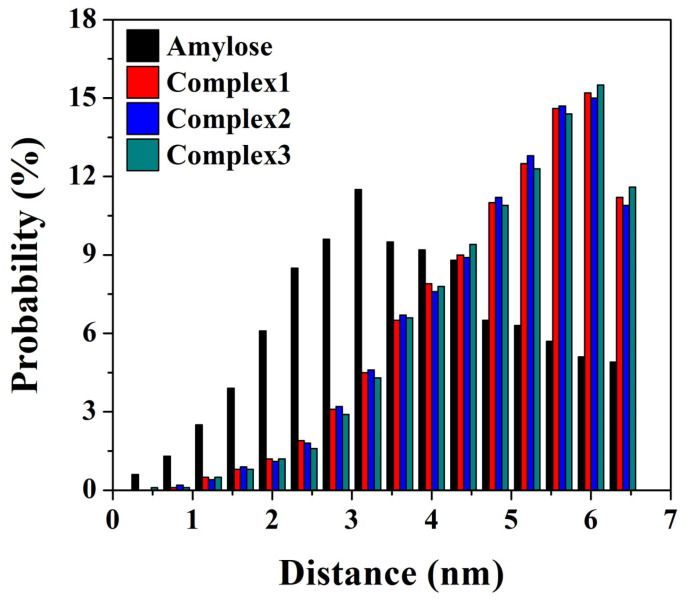
The probability distribution of the distances of two ends of amylose molecule in both the single-strand system (black bars) and three amylose–zein complexes with different initial values (red, blue and green bars).

**Figure 5 foods-12-01418-f005:**
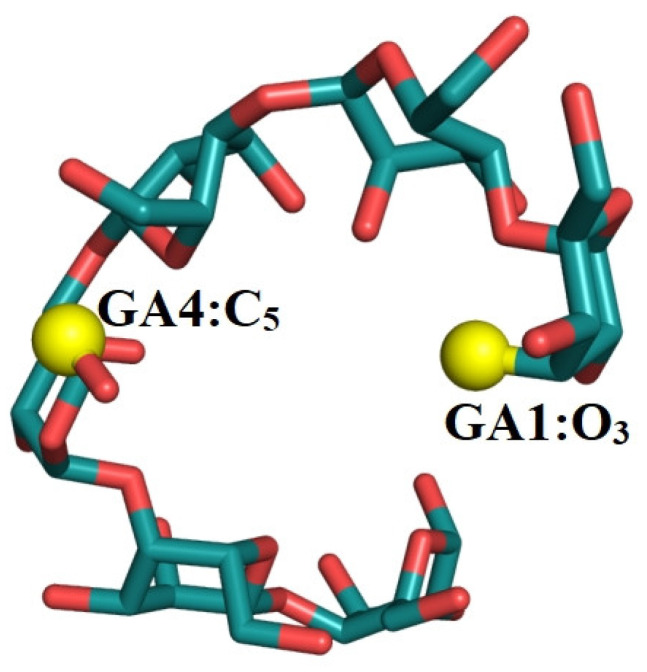
Atom distance between the O_3_ atom in the residue x and the C_5_ atom in the residue x+3, the two yellow atoms are O_3_ and C_5_, respectively.

**Figure 6 foods-12-01418-f006:**
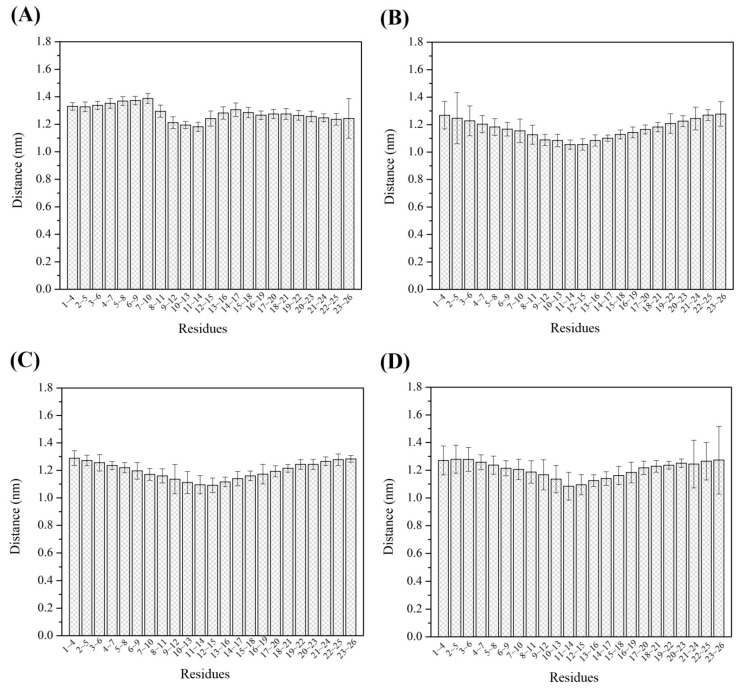
Atom distance of single amylose molecule (**A**), complex1 (**B**), complex2 (**C**), and complex3 (**D**).

**Figure 7 foods-12-01418-f007:**
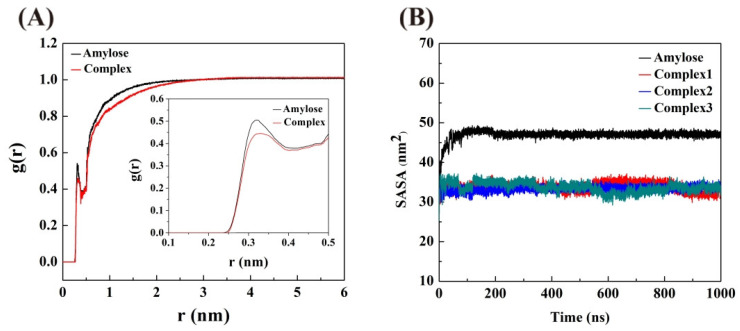
(**A**) Radial distribution functions (RDF) of amylose molecule in single-strand system (black lines) and the average RDF of amylose molecule in three different amylose–zein complexes with different initial values (red lines); (**B**) Solvent accessible surface area (SASA) of amylose molecule in both the single-strand system (black line) and three amylose–zein complexes with different initial values (red, blue and green lines).

**Figure 8 foods-12-01418-f008:**
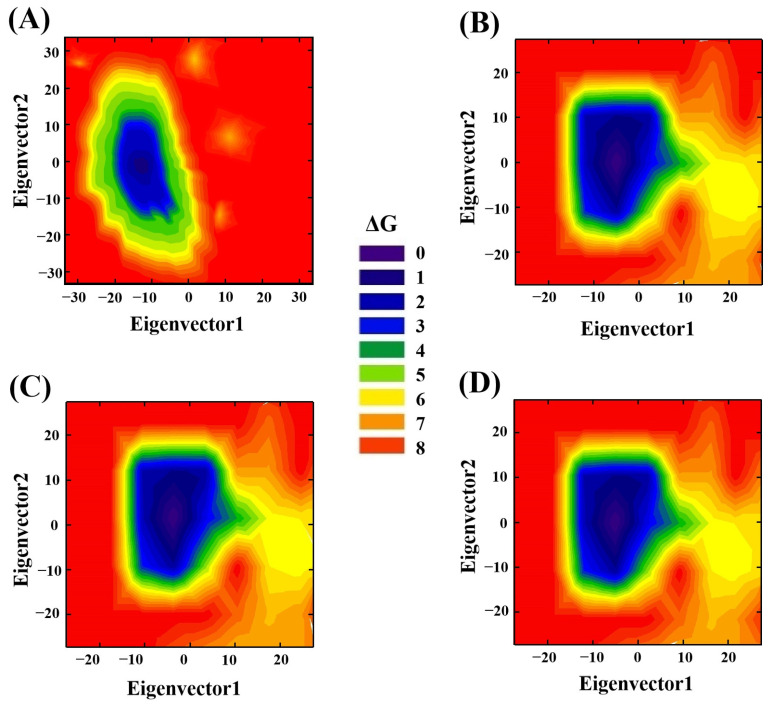
Principal component analysis (PCA) of single amylose molecule (**A**), complex1 (**B**), complex2 (**C**), and complex3 (**D**). The different colors in the figure legend represent different levels of energy.

**Figure 9 foods-12-01418-f009:**
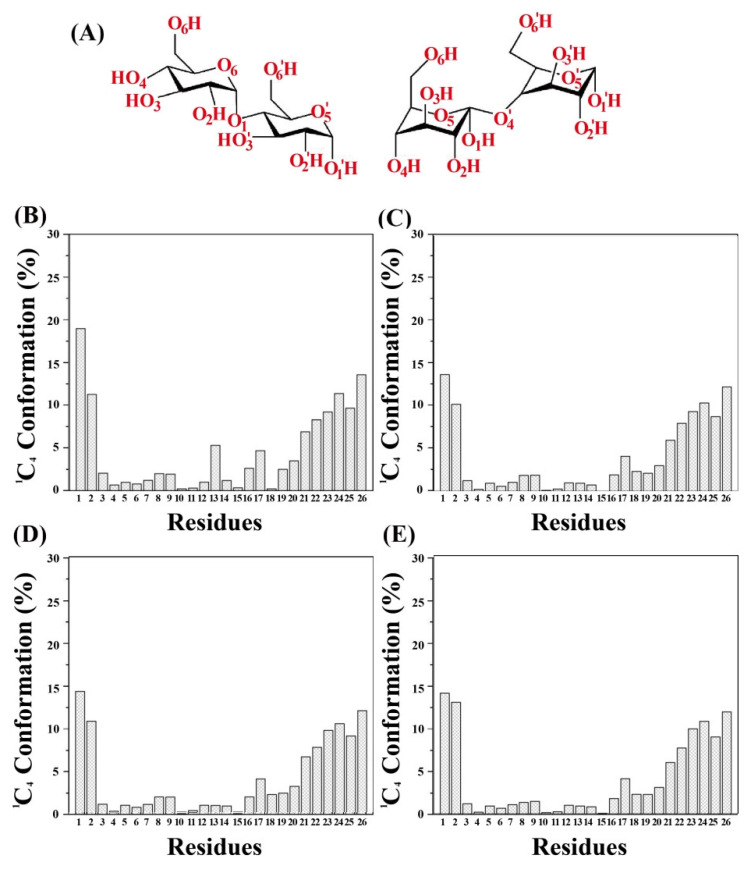
(**A**) Conformational structures of ^4^C_1_ (left) and ^1^C_4_ (right); The distribution proportion of ^1^C_4_ chair conformation of glucose residues of amylose molecule in both the single-strand system (**B**), complex1 (**C**), complex2 (**D**), and complex3 (**E**).

**Table 1 foods-12-01418-t001:** The number of O_2_-O_3_ intramolecular hydrogen bonds of amylose molecule in single-strand system and three amylose–zein complexes with different initial values.

Time (ns)	The Number of O_2_-O_3_ Intramolecular Hydrogen Bonds			
	Amylose	Complex1	Complex2	Complex3
0	23	23	23	23
0–200	5.42 ± 0.51	16.03 ± 1.22	15.72 ± 0.93	15.54 ± 0.82
200–400	5.58 ± 0.62	15.34 ± 1.03	14.84 ± 1.12	14.73 ± 1.01
400–600	5.31 ± 0.42	13.52 ± 0.81	12.93 ± 0.82	13.74 ± 0.93
600–800	5.17 ± 0.31	12.44 ± 0.92	11.89 ± 0.91	12.13 ± 0.82
800–1000	4.73 ± 0.55	11.53 ± 1.12	11.14 ± 0.92	10.81 ± 1.22

The values are expressed as mean ± standard deviation of three replicates.

**Table 2 foods-12-01418-t002:** The number of O_6_-O_2_/O_3_ intramolecular hydrogen bonds of amylose molecule in single-strand system and three amylose–zein complexes with different initial values.

Time (ns)	The Number of O_6_-O_2_/O_3_ Intramolecular Hydrogen Bonds			
	Amylose	Complex1	Complex2	Complex3
0	15	15	15	15
0–0.25	6.12 ± 1.43	11.63 ± 1.21	11.93 ± 1.31	11.24 ± 0.91
0.25–0.5	6.82 ± 1.51	10.14 ± 1.01	10.92 ± 0.92	10.43 ± 1.12
0.5–0.75	6.64 ± 1.81	9.24 ± 0.91	9.74 ± 1.11	9.54 ± 0.82
0.75–1.0	4.23 ± 1.62	7.83 ± 1.11	8.03 ± 1.21	8.23 ± 0.91
1.0–1.5	3.83 ± 1.11	6.44 ± 0.72	7.62 ± 0.83	7.14 ± 0.92
1.5–2.0	2.14 ± 0.61	6.53 ± 0.91	6.92 ± 1.12	6.23 ± 1.21
2.0–3.0	0.83 ± 0.31	6.13 ± 1.22	5.94 ± 1.25	6.84 ± 1.33
3.0–4.0	0	6.82 ± 1.13	6.53 ± 1.12	6.44 ± 1.22
4.0–100	0	5.54 ± 0.93	6.24 ± 1.21	5.83 ± 0.91
100–500	0	4.93 ± 0.81	5.53 ± 1.31	5.33 ± 1.12
500–1000	0	3.92 ± 0.93	4.72 ± 1.14	4.84 ± 0.77

The values are expressed as mean ± standard deviation of three replicates.

## Data Availability

The data are available from the corresponding author.

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
