# Peer review of "The Mechanism Underlying the Amylose-Zein Complexation Process and the Stability of the Molecular Conformation of Amylose-Zein Complexes in Water Based on Molecular Dynamics Simulation"

_foods, 2023, doi:10.3390/foods12071418_

Round 1

Reviewer 1 Report

The Authors recently obtained starch-zein core-shell microparticles with high slowly digestible starch and resistant starch contents. Presented paper is a continuation of these studies.  The Authors used molecular dynamics simulations to elucidate the mechanism involved in amylose-zein complexation and the stability of the molecular conformation of amylose-zein complexes in water at the atomic and molecular levels. The research suggested a significantly higher conformational stability for amylose-zein complexes than for amylose in water. They found, that the amylose-zein interaction inhibited water permeation into the amylose cavity. This research showed decreased replacement of amylose intramolecular hydrogen bonds with amylose-water bonds and an increased replacement of amylose-water bonds with amylose-zein bonds. Investigated amylose in amylose-zein complexes displayed the thermodynamically stable 4C1 conformation. This research is important, as it concerns developing novel starch-protein products with low digestibility. English level is good. The paper is well organized. Discussion is very interesting.

Author Response

Response to Reviewer 1 Comments:

Thank you for taking the time to read this manuscript. Thanks for your positive comments.

Reviewer 2 Report

The study elucidate the mechanism  involved in amylose-zein complexation and the stability of the molecular conformation of amylose-zein complexes in water at the atomic and molecular levels.

The work is very complete, with a series of experiments that meet the requirements of the journal. The applied methodology is correct however some corrections are necessary:

1. Please modify abstract in a more concise way;

2. Clarify better the aim of the work at the end of introduction section;

3. Some details are necessary in material paragraph.

4. Statistical analysis is missed. Please add;

5. Improve the quality of figures are not easy for readers;

6. In general, improve work discussions.

7. Conclusion shortened, results should not be repeated in the conclusions, neither the objective of the work.

Author Response

Response to Reviewer 2 Comments:

Point 1: The study elucidate the mechanism  involved in amylose-zein complexation and the stability of the molecular conformation of amylose-zein complexes in water at the atomic and molecular levels. The work is very complete, with a series of experiments that meet the requirements of the journal. The applied methodology is correct however some corrections are necessary.

Response 1: Thanks for your positive comments. The raised questions are answered point by point below.

Point 2: Please modify abstract in a more concise way.

Response 2: Thanks for this good advice. We have modified abstract in a more concise way and marked it in red (now line 18–32).

Point 3: Clarify better the aim of the work at the end of introduction section.

Response 3: Thanks for this good advice. We clarified better the aim of the work about the relevance of these findings for the food research at the end of introduction section and marked it in red (now line 87–98, line 99–101, line 112–119).

Point 4: Some details are necessary in material paragraph.

Response 4: Thanks for this good advice. We apologize for making the mistakes. We have added some details in material paragraph and marked it in red (now line 122–124).

Point 5: Statistical analysis is missed. Please add

Response 5: Thanks for this good advice. We apologize for making the mistakes. We have added statistical analysis and marked it in red (now line 154–157).

Point 6: Improve the quality of figures are not easy for readers.

Response 6: Thanks for this good advice. We apologize for making the mistakes. We have improved the quality of figures (now Figure 2, Figure 3, Figure 4, Figure 5, Figure 6, and Figure 7). We have revised the explanatory title of Figure 2, Figure 3, and Figure 6 and marked it in red (now line 195–201, line 279–282, line 429–430).

Point 7: In general, improve work discussions.

Response 7: Thanks for this good advice. We have improved work discussions and marked it in red (now line 176–181, line 293–295, line 316–319, line 354–357, line 413–416).

Point 8: Conclusion shortened, results should not be repeated in the conclusions, neither the objective of the work.

Response 8: Thanks for this good advice. We have shortened and reviesd conclusion. The revised parts are marked in red (now line 479–496).

Reviewer 3 Report

The manuscript is interesting. However, some points are needed to be addressed.

I would suggest adding in the abstract some information about the relevance of these findings for the food research.

Line 41: specify the rapidly digestible starch is…

Line 67: a reference is needed.

Line 91: Please, specify why the amylose-zein complexation process in water and the thermodynamically stable con-90 figuration of the amylose-zein complexes are still not fully understood

2.1 Materials: add the specifications of the software used.

Line 144: this process (starch gelatinization) happens at high temperature, not at room temperature. Specify better this point, also in the whole discussion. Temperature plays an important role in starch reactions in water, so the authors need to better clarify this aspect in the whole discussion.

3.3: Use Root mean square deviation alongside the acronym in the title.

3.4: Use radius of gyration alongside the acronym in the title

3.5. Use Root mean square fluctuation alongside the acronym in the title

3.7. Use Radial distribution functions alongside the acronym in the title… the same for 2.8 and 3.9

Overall, it would be useful to have a description of the practical implications of such findings for the food development for example, or for the nutritional value of the products.

A check of the English in the manuscript is needed, there are some mistakes in the sentences structure.

Author Response

Response to Reviewer 3 Comments:

Point 1: The manuscript is interesting. However, some points are needed to be addressed.

Response 1: Thanks for your positive comments. The raised questions are answered point by point below.

Point 2: I would suggest adding in the abstract some information about the relevance of these findings for the food research. 

Response 2: Thanks for this good advice. We have added in the abstract some information about the relevance of these findings for the food research and marked it in red (now line 28–32). In addition, we have clarified better the aim of the work about the relevance of these findings for the food research at the end of introduction section and marked it in red (now line 87–98, line 99–101, line 112–119).

Point 3: Line 41: specify the rapidly digestible starch is…

Response 3: Thanks for this good advice. We have specified the rapidly digestible starch and marked it in red (now line 37–40).

Point 4: Line 67: a reference is needed.

Response 4: We have added references and marked them in red (now line 64–66).

Point 5: Line 91: Please, specify why the amylose-zein complexation process in water and the thermodynamically stable figuration of the amylose-zein complexes are still not fully understood.

Response 5: Thanks for this good advice. We have deleted this sentence (now line 86).

Point 6: 2.1 Materials: add the specifications of the software used.

Response 6: Thanks for this good advice. We have added the specifications of the software used and marked it in red (now line 122–124).

Point 7: Line 144: this process (starch gelatinization) happens at high temperature, not at room temperature. Specify better this point, also in the whole discussion. Temperature plays an important role in starch reactions in water, so the authors need to better clarify this aspect in the whole discussion.

Response 7: Thanks for this good advice. We have clarified this aspect in line 87–101.

Point 8: 3.3: Use Root mean square deviation alongside the acronym in the title.

Response 8: Thanks for this good advice. We apologize for making the mistakes. We have replaced "RMSD" with "Root mean square deviation (RMSD)" and marked it in red (now line 256). 

Point 9: 3.4: Use radius of gyration alongside the acronym in the title.

Response 9: Thanks for this good advice. We apologize for making the mistakes. We have replaced "Rg" with "Radius of gyration (Rg)" and marked it in red (now line 283).

Point 10: 3.5. Use Root mean square fluctuation alongside the acronym in the title.

Response 10: Thanks for this good advice. We apologize for making the mistakes. We have replaced "RMSF" with "Root mean square fluctuation (RMSF)" and marked it in red (now line 296).

Point 11: 3.7. Use Radial distribution functions alongside the acronym in the title… the same for 2.8 and 3.9.

Response 11: Thanks for this good advice. We apologize for making the mistakes. We have revised these titles and marked it in red (now line 346, line 372, line 395).

Point 12: Overall, it would be useful to have a description of the practical implications of such findings for the food development for example, or for the nutritional value of the products.

Response 12: Thanks for this good advice. We have described the practical implications of such findings for the food development and the nutritional value of the products at the end of introduction section. The revised parts are marked in red (now line 87–98, line 99–101, line 112–119).

Point 13: A check of the English in the manuscript is needed, there are some mistakes in the sentences structure.

Response 13: Thanks for this good advice. We apologize for making the mistakes. We have asked English-speaking colleague to corrected the sentences structure. The revised parts are marked in red (now line 18–20, line 52–54, line 293–295, line 316–319, line 354–357).

Reviewer 4 Report

The authors investigated the molecular mechanism and stability of amylose-zein complexes using molecular dynamics simulations. The results are applicable as a guide for developing low-digestibility starch-protein products in starch processing. This study is well-organized and written. The conclusions also justify the set objectives.

Authors should check the citation in Line 148. An error in citation ….amylose-ligand complexes exhibit a left-handed V-helical structure [28]. This is from Valletta et al., 1964. In addition, this should be updated with a recent citation since it’s not a method.

Author Response

Response to Reviewer 4 Comments:

Point 1: The authors investigated the molecular mechanism and stability of amylose-zein complexes using molecular dynamics simulations. The results are applicable as a guide for developing low-digestibility starch-protein products in starch processing. This study is well-organized and written. The conclusions also justify the set objectives.

Response 1: Thanks for your positive comments. The raised questions are answered point by point below.

Point 2: Authors should check the citation in Line 148. An error in citation ….amylose-ligand complexes exhibit a left-handed V-helical structure [28]. This is from Valletta et al., 1964. In addition, this should be updated with a recent citation since it’s not a method.

Response 2: Thanks for this good advice. We apologize for making the mistakes. We have replaced "Valletta, R.M. ..." with a recent citation about complexation process of amylose-ligand complexes in line 564–565 and marked it in red.

Round 2

Reviewer 2 Report

I have no additional suggestion.

Author Response

Thank you for taking the time to read this manuscript. Thanks for your positive comments.